# Utility of Lymphadenectomy in Prostate Cancer: Where Do We Stand?

**DOI:** 10.3390/jcm11092343

**Published:** 2022-04-22

**Authors:** Bartosz Małkiewicz, Paweł Kiełb, Jakub Karwacki, Róża Czerwińska, Paulina Długosz, Artur Lemiński, Łukasz Nowak, Wojciech Krajewski, Tomasz Szydełko

**Affiliations:** 1University Center of Excellence in Urology, Department of Minimally Invasive and Robotic Urology, Wroclaw Medical University, 50-556 Wroclaw, Poland; pawel.kielb@student.umw.edu.pl (P.K.); jakub.karwacki@student.umw.edu.pl (J.K.); roza.czerwinska@student.umw.edu.pl (R.C.); paulina.dlugosz@student.umw.edu.pl (P.D.); lukasz.nowak@student.umw.edu.pl (Ł.N.); wojciech.krajewski@umw.edu.pl (W.K.); tomasz.szydelko@umw.edu.pl (T.S.); 2Department of Urology and Urological Oncology, Pomeranian Medical University, Powstańców Wielkopolskich 72, 70-111 Szczecin, Poland; artur.leminski@pum.edu.pl

**Keywords:** prostate cancer, lymph node dissection, lymphadenectomy, radical prostatectomy

## Abstract

The purpose of this review is to summarize the current knowledge on lymph node dissection (LND) in prostate cancer (PCa) patients undergoing radical prostatectomy (RP). Despite a growing body of evidence, the utility and therapeutic and prognostic value of such an approach, as well as the optimal extent of LND, remain unsolved issues. Although LND is the most accurate staging procedure, the direct therapeutic effect is still not evident from the current literature, which limits the possibility of establishing clear recommendations. This indicates the need for further robust and adequately designed high-quality clinical trials.

## 1. Introduction

Prostate cancer is the second most common cancer in men (after lung cancer) and the fifth leading cause of death worldwide [1]. One man in eight is going to be diagnosed with PCa. Lymph node metastases constitute a poor prognostic factor for patients with PCa, both in terms of biochemical recurrence (BCR) and survival [2]. Although nowadays LND can be avoided more frequently by using various nomograms assessing the probability of lymph node invasion (LNI), lymphadenectomy performed during prostatectomy remains the first-choice procedure for evaluating metastasis presence [3]. Despite LND being an excellent staging tool, and some studies having indicated its positive effect on BCR-free survival, the overall therapeutic benefit of LND is questionable and unclear [4]. Moreover, this surgical approach entails an increased risk of peri- and postoperative complications, longer operative time, and increased morbidity. According to current guidelines, PLND (pelvic LND) should be performed especially among patients with high-risk and intermediate-risk PCa when the probability of LNI exceeds 5% [5]. In this review, we aim to establish the benefits and harms of the present-day approach, describe the role of PLND in the management of PCa and seek future possibilities.

## 2. Evidence Acquisition

For the purposes of this narrative review, we conducted a comprehensive English language literature research for original and review articles using the Medline database and grey literature through March 2022. We searched for the combination of following terms: prostate cancer; lymph node dissection; lymphadenectomy; radical prostatectomy. We found 1103 related articles, and the final number of papers selected for this manuscript was 178. Studies with the highest level of evidence and relevance to the discussed topics (129) were selected, with the consensus of the authors.

## 3. Diagnosis and Prediction

The stage of the cancer is described by the TNM classification, where T stands for the progression of the disease within the prostate gland, N, lymph node metastases, and M, distant metastases. The TNM-confirmed extent of PCa is crucial for planning treatment strategies.

There are several diagnostic methods used to determine whether LNI or distant organ metastases occur. Imaging procedures play an important complementary role in the primary detection, staging, post-treatment assessment, and recurrence of prostate cancer.

Computed tomography (CT) and magnetic resonance imaging (MRI) are the conventional imaging techniques, confirmatory of the shape and size of the nodes. Although they might be helpful in detecting metastases, their efficiency is low and may misdirect the patient’s therapy [6,7]. One of the meta-analyses shows not statistically significant but comparable poor performance of these two methods. Pooled sensitivity for CT was 0.42, and for MRI, 0.39. In the case of pooled specificity the result was 0.82 for both diagnostic methods [6].

Currently, MRI imaging offers more advanced procedures. One of them is diffusion-weighted imaging (DWI). DWI sequences show the Brownian movements of water molecules. MRI DWI lymph node (LN) staging has low sensitivity but high specificity and performs better than standard MRI imaging [8,9]. One study reports the sensitivity of DWI-MRI in detection of lymph node invasion at the level of 41% and specificity at the level of 94% [7].

Positron emission tomography (PET) imaging in LNI found a few different radiotracers that can be used in the field of PCa. The most thoroughly tested substances are radiolabeled glucose, choline, fluciclovine, acetate or NaF [10].

The scope of application of 18F-Fluorodeoxyglucose (18F-FDG) is very limited, yet can be used to detect LNI in some cases [7,11,12,13]. For example, Jadvar (2016) points out that FDG can only be useful in the detection and staging of high-grade tumors (Gleason score > 7). Furthermore, FDG PET-CT detected metastatic disease in LN and/or bone only in six of the nine (67%) patients [12].

Globally, two choline derivatives are used—radiolabeled with Carbone-11 (11C) or Fluor-18 (18F). The strength of this method lies in the possibility of detecting LNI, bones and distant organs’ metastases; however, the detection rate is low. Fraum et al. (2018) point out that 11C choline has sensitivities for nodal metastatic disease of 60% in the case of a per-patient basis and 41% in the per-node basis [11]. This method gains diagnostic accuracy when prostate-specific antigen (PSA) serum level is high enough. Therefore, according to the European Association of Urology (the EAU), it is recommended to use choline-based PET imaging in BCR patients after RP if their PSA serum level is ≥1 ng/mL [5,7,11,14].

Another radiotracer used in PET Imaging is acetate, which seems to be better than choline at detecting local recurrences and LN metastases [10]. Some researchers point out that its sensitivity is rather unsatisfactory and the methods using this substance as a tracer have several limitations, for example, the minimum detectable tumor size of 5 mm, which is an important constraint [15].

The imaging methods are under constant improvement, and new substances are being tested. Another interesting PET/CT method uses a protein called PSMA—prostate-specific membrane antigen. PSMA is physiologically expressed by prostate cells and overexpressed in PCa cells, as well as some other malignant tissues, and it is presumed to be a valuable metastasis marker [11,16]. Derivative radiolabeled tumor targeting molecules were created—68Ga-PSMA and 18F-DCFPyL. These FDA-approved radioisotope-bound proteins show promising clinical potential [17]. They seem to be helpful in initial staging as well as detecting recurrences or even treatment assessment [11,18,19,20]. One meta-analysis indicates that, in retrospective studies, the method using 68Ga achieved widely varying sensitivity and specificity (33.3% to 100%). The detection rate of 68-Ga-PSMA PET in patients with BCR after RP in the PSA subgroups <0.2 ng/mL, 0.2–0.49 ng/mL and 0.5 to <1.0 ng/mL ranged from 11.3–50%, 20–72.7% and 25–87.5%, respectively [18].

Another discovery related to PSMA pertains to its similarity to the N-acetyl-aspartyl-glutamate peptidase (NAAALDASE). Metastasis detection is possible using inhibitors targeting the expressed PSMA. One example is a small molecule inhibitor 99 mTc labelled MIP-1404. Research indicates its potential for finding LNI and soft tissue or bone metastases [16,19,21]. However, there are significant limitations to the use of this method. Primarily, researchers point out that PSMA uptake is not specific for PCa but also characterizes many benign tissues. Moreover, the inhibition of PSMA expression is common in advanced stages of the disease and up to 10% of PCa cases do not overexpress this protein. The same meta-analyses reported high specificity at the level of 95% but a poor sensitivity at the level of 49% in primary nodal staging [20].

In conclusion, none of the imaging methods are efficient enough to be considered as a gold standard. All of them have strengths and weaknesses, although the simultaneous use of several methods may be useful in improving the accuracy of detection. They might be helpful when it comes to optimizing treatment and localizing recurrences. Currently, even highly developed imaging techniques are not sufficient to fully replace PLND; therefore, more studies are required [6,7,9,10].

Due to the fact that lymphadenectomy is an invasive staging procedure, and markers of PCa metastases are difficult to be interpreted as single parameters, several nomograms have been created to enhance decision making and establish estimated the probability of LNI, etc. (e.g., positive margins or extra-capsular extension) [22]. Most of the PCa-related nomograms can be divided into diagnostic, post-diagnostic and before- or after-treatment tools. The nomograms predicting LNI are before-treatment assessment tools and are widely used to facilitate decision making about whether to apply PLND during RP or not [23].

The Briganti nomogram, one of the most widely used nomograms predicting LNI in PCa, is based on serum PSA levels, clinical T-stage, primary and secondary Gleason grades and percentage of positive cores [24]. It suggests performing PLND if the calculated risk is higher than or equal to 5%. Its internal validation evaluated the accuracy of prediction at the level of 87.6%; LNI would be missed in 1.5%. The 2018 Briganti nomogram (also known as the Gandaglia nomogram) is a model that predicts LNI in patients diagnosed with MRI-targeted and systematic biopsies. It considers PSA levels, clinical stage at multi-parametric MRI, maximum lesion diameter and biopsy results; the suggested cutoff is 7% [25]. Other nomograms used in anticipating LNI are the Partin tables and Memorial Sloan Kettering Cancer Center (MSKCC) nomogram. They are based on TNM, preoperative PSA level and biopsy Gleason score [26,27,28].

There are more tools, such as Godoy nomogram, Roach formula or CAPRA score, used, though many of these lack external validations. Briganti, Partin and MSKCC nomograms have similar prediction accuracy and (along with the Roach formula) are recommended by the EAU for preoperative LNI risk assessment [29,30]. All the abovementioned nomograms can be excellent tools for predicting nodal involvement, although it is crucial to understand that the treated population should be similar to the population on which a certain nomogram-associated study was conducted [31]. These assessment instruments are not meant to define the treatment strategy, but to help clinicians make proper decisions, and a decision-making process should always be multifactorial. Furthermore, a great number of nomograms evaluating LNI may also be an issue. A meta-nomogram compiling different predictive tools would play a pivotal role in the PLND strategy in PCa.

## 4. Anatomical Extent of PLND

Prostate cancer disseminates through venous routes, peri-neural spaces and the lymphatic network. The main causes of lymphadenopathy in prostate cancer are: metastases, hyperplastic and regressive alterations [32].

The goal of PLND is to remove lymph nodes and lymphatic vessels/trunks from the landing zones for metastases. Prostate cancer is originally distributed to regional LNs [33]. The first site to which lymph flow carries cancer cells is known as a sentinel node. According to this theory, the presence of metastases in sentinel nodes can suggest that they are present in other LNs, and similarly the lack thereof suggests that other LNs are cancer-free. It reflects disease progression in some cancers well, although its role in deeply located cancers (such as prostate cancer) requires further investigation, with a lack of concrete data [34]. Diagnostic value and significance of the sentinel node technique will be discussed in detail in Section 8 of the review.

Earlier, it was believed that the primary landing site consists of obturator, internal and external iliac LNs [35]. Researchers concluded that patterns of dissemination and drainage for prostatic glands are not identical, but data are too limited [36]. As proven by isotope-based studies, metastatic cells do not sequentially spread, but rather can be detected all the way to the inferior mesenteric artery area. A multimodal mapping study conducted by Mattei et al., discredits this theory—in 34 patients who underwent RP for biopsy-confirmed cN0cM0 prostate cancer, preoperative single-photon emission computed tomography (SPECT)/CT and intraoperative gamma probe were used after the injection of technetium-99 m into the prostate gland. Positive nodes were detected along the external iliac vessels and obturator fossa (38%), internal iliac vessels (25%), common iliac vessels (16%), perirectal and pre-sacral area (8%) and the para-aortic/para-caval (12%) and inguinal regions (1%) [33]. Another study, a SPECT-based virtual 3D atlas of the landing sites, demonstrated sentinel nodes present in 61 high-risk patients who underwent PLND and RP [37]. Furthermore, some studies suggest that a larger positive lymph node can interfere with lymphatic flow [38,39].

In recent years, attention has been paid to the anatomical region called Marcille’s fossa (limited by the ala of the sacrum, the medial border of the psoas muscle and the anterolateral side of the fifth lumbar vertebra/promontory, covered by the iliac vessels). It was proven to be connected with the prostatic lymphatic system and linked with high metastatic load involvement—positive Marcille’s nodes are correlated with metastases in other locations in high-risk prostate cancer patients [40]. Marcille’s triangle can only be accessed by full exposure, medial retraction and mobilization of the external iliac vessels along with the ureter [40]. At the moment, Marcille’s lymphadenectomy, also known as “marcillectomy”, is not recommended as a standard procedure, as there are no prediction factors yet available.

The main objective of PLND is to find out the loco-regional extent of cancer, the risk of progression or recurrence, and to determine if therapy is needed [41,42]. Secondly, it can be a form of treatment for patients who already have already undergone local therapy to get rid of leftover tumor [43]. Therefore, it is necessary to study the overall lymphatic drainage pattern for the prostate [44].

The prostatic lymphatic drainage mostly occurs in the cephalad direction, following the blood supply route of the organ. It incorporates the external and internal iliac artery areas and the obturator fossa. According to other anatomic studies, there are ascending flow (which drains the cranial prostate into external iliac LN), lateral flow (draining into the hypo-gastric node chain) and posterior flow (draining lymph from the caudal prostate into the sub-aortic LN of the sacral promontory) duct groups [45,46]. Surgical studies showed limited usefulness, as they can only evaluate exposed and removed LN—in areas spared in lymphadenectomy, it is impossible to know if there are positive nodes [47]. Bayer et al., conducted an embryological study arguing that the knowledge of the ontogenesis of the contents of pelvis compartments is crucial for the ability to propose suitable and optimal PLND templates [48].

The template for LND is defined by the localization of metastases and the lymphatic drainage pattern of a particular cancer, as well as the state of the primary lesion. Regarding the theoretical anatomical extent of the procedure, we can divide it into the following types: limited, standard, extended and super-extended (Figure 1). As of today, there is no standardization proposed. There seems to be consensus neither as to what is the optimal extent for each type of procedure, nor to what terminology should be used. It does not help that, in some research papers, a unique take is employed, or even that sometimes the extent is not specified at all. Creating a standardized nomenclature is desired as it would allow for better regularity in practice between institutions and would make it easier to accurately compare results for future studies on the subject. The Committee on Classification of Regional Lymph Nodes of the Japan Society of Clinical Oncology created guidelines aiming to help overcome these drawbacks and should be used in future research [48,49].

The limited PLND (lPLND) engages the obturator fossa area LN only (located medio-caudally to the external iliac vein, atop the tendinous arch of the levator ani muscle and internal obturator muscle), while the standard PLND covers the obturator and external iliac nodes (proximally located along or between external iliac vessels, distally next to the deep inguinal ring, crossed by the deep circumflex iliac vessels) [50,51].

The extended PLND (ePLND) consists of lymph node groups covered by the standard procedure as well as additional ones, such as hypo-gastric, pre-sacral (along the sacral concavity), internal, and common iliac nodes (stretching on the common iliac vessels before the aortic bifurcation) [51,52]. Salvage extended PLND is recommended in recurrent prostate cancer, with the additional removal of inter-iliac and para-aortic LN [48,53]. The total number of nodes removed is crucial to maintain the accuracy of the staging procedure. The higher the number, the greater the chances of detecting a node-positive case [54]. Based on research led by Weingärtner et al., a mean LN yield of 20 was suggested as a sufficient PLND guideline [32]. It is recommended by the EAU and the National Comprehensive Cancer Network (NCCN) to adapt PLND usage with the help of nomograms and other risk stratification tools in order to predict lymph node metastases preoperatively [55]. This allows for the disqualification of low-risk PCa patients, in whom the probability of being node-positive is <3% [35,56]. It is worth noting that these tools might require a revisit because they were developed predominantly on data collected from lPLND performed in older patient series [52].

Substantial debate has taken place as to what the proper boundaries of PLND ought to be. In clinical practice, it is usually enough to dissect only the obturator LN due to the reportedly relatively low total of node-positive cases (<8%), with only 25% of total positive nodes found surrounding the internal iliac artery [30,50,52,57,58]. Pre-sacral and common iliac LN metastases are uncommon [35]. High diagnostic staging accuracy was presented for standard PLND [30,59]. On the contrary, the addition of more nodal areas improves survival in pN0 in patients, effectively by the elimination of micro-metastases [40].

An attempt was made to determine patterns of prostate dissemination regarding dominant tumor mass location. It was discovered in various studies that 10–46% of positive LNs were located contra-laterally, with only a 10–17% rate of contralateral-only cases, with false predictive rates of 14–29% [60]. Based on those results, some authors suggest that the only reliable lymph node staging method is complete bilateral lymphadenectomy [61,62].

## 5. Oncological Outcomes

There are several approaches to treating patients with localized PCa. According to researchers, the efficacy of radical prostatectomy with ePLND is more beneficial than just radiotherapy itself [63]. The ePLND is not only a reliable tumor staging tool but may also have a potential therapeutic effect; however, this is not explicit [41,64,65].

Clinical recurrence can be evaluated by examining the presence of distant metastasis. Some studies show that patients who underwent PLND present a higher risk of recurrence. However, in low-risk PCa patients, no recurrences have been observed [41]. No available studies have provided any relevant survival rate (both cancer-specific and overall mortality) difference between the PLND and non-PLND treatment [41]. An additional parameter taken into consideration is the increase in PSA level, which can indicate the BCR [66].

Taking into consideration clinical recurrences, there were no studies that reported on the difference in distant metastasis between sPLND and ePLND procedures [41,64]. There are trials that seek to determine whether lPLND or ePLND for PCa have better oncological outcomes.

Lestingi et al., investigate a prospective randomized phase 3 trial in a total of 300 patients with intermediate- or high-risk clinically localized PCa [64]. The group was split into two halves, one having ePLND and one having lPLND carried out. They found that extended removal of LN did not reduce the BCR of PCa in the expected range. The median biochemical relapse-free survival (BRFS) was 61.4 mo in the lPLND group and was not reached in the ePLND group (hazard ratio (HR) 0.91, 95% confidence interval (CI) 0.63–1.32; *p* = 0.6). Median metastasis-free survival (MFS) was not reached in either group (HR 0.57, 95% CI 0.17–1.8; *p* = 0.3). In summary, the differences in early oncological outcomes were not demonstrated [64].

The second trial provided by Touijer et al., is a single-center randomized trial in a total of 1440 patients assigned to limited or extended PLND [67]. A total of 700 were randomized to lPLND and 740 to ePLND. In this clinical trial, a difference was not found in the rate of biochemical recurrence of prostate cancer between the two procedures. The median number of nodes retrieved was 12 (interquartile range [IQR] 8–17) for lPLND and 14 (IQR 10–20) for ePLND; the corresponding rate of positive nodes was 12% and 14% (difference 1.9%, 95% confidence interval [CI] 5.4% to 1.5%; *p* = 0.3). With a median follow-up of 3.1 years, there was no significant difference in the rate of biochemical recurrence between the groups (hazard ratio 1.04, 95% CI 0.93–1.15; *p* = 0.5) [67]. Extended PLND did not improve the chances of BCR-free outcome for men with clinically localized PCa over lPLND. Moreover, the observed difference both in nodal count and the rate of positive nodes between the two templates was lower than expected [67]. Subsequent trials comparing those methods are still recommended.

Despite the drawbacks, performing the ePLND still appears to be warranted. It allows the assessment of the cancer spread, including micro-metastases not detectable by imaging techniques [68]. As a cancer staging tool, ePLND helps in a non-direct manner to enhance the oncological outcome [41]. This argument would need to be examined in subsequent research.

Considering the available data, the therapeutic role and oncologic efficacy prospects of PLND remain unclear. There is no evidence to back up the claim that PLND improves oncological outcomes over no PLND. Only some particular subgroups of the patients might benefit from the procedure. In addition, weighing non-oncological outcomes, performing PLND was associated with a higher risk of intraoperative and perioperative complications; however, there was no evidence showing difference in functional outcomes, such as erectile function and urinary incontinence.

Since there are still controversies associated with this procedure, further clinical trials are required [41,64,65]. The shortage of solid evidence should lead to individual patient eligibility for surgery or disqualification. Personal risk ought to be taken into consideration and patients, therefore, judiciously selected. As long as we do not have certain trial results, clinicians ought to follow the recommendations of the EAU guideline and perform ePLND for PCa patients who present more than 5% risk of LNI [5].

## 6. Complications of PLND

PLND is a procedure bearing relatively both short- and long-term complications and mortality rates, resting at a 20–35% overall complication rate, and mortality of under 1%. None of the available studies showed any relevant survival rate difference (both cancer-specific and for overall mortality) between PLND and no PLND treatment [51].

Postoperative complications remain in relation to dissection template extent—more invasive procedures lead to increased postoperative organ impairment [34,69]. Among the most common minor complaints we can name are: wound infection (<5% of patients), atelectasis, small bowel obstruction (<2% of patients), and ureteral and vascular injuries (<1%, usually recognized and fixed at the time of the original operation) [70]. Lymphoceles are positively linked with a greater dissection template. During surgical lymphadenectomy, both afferent and efferent lymphatic vessels are susceptible to thermal or mechanical injury, more likely to occur by blunt dissection or gross plucking (on the contrary, en bloc harvesting reduces the probability of these complications) [48]. They are extremely common, yet unlikely to be symptomatic or cause morbidity. Patients with lymphoceles presented higher rates of deep venous thrombosis and pulmonary embolism [69,71]. When it comes to severe complications, the most serious ones are pulmonary insufficiency, chylous ascites (lymphatic leak) and lymphatic cysts, lymphatic fistula or chylo-pelvic fistula [48,72,73]. Thromboembolic complications are sufficiently rare and in the majority of cases (over 99%) do not require treatment [71]. Routine pharmacological prophylaxis is currently recommended to be considered for intermediate and high thromboembolic risk indicated for ePLND. Mechanical prophylaxis, however, is recommended for all PLND patients [71]. Patients with deep venous thrombosis presented increased pulmonary embolism risk and were more likely to be re-operated upon [69].

Intra-operatively, bleeding may occur, especially from damaged aorta, vena cava or iliac blood vessels. Serious bleeding may require blood transfer. This adverse effect frequency is related to the operating time and the surgeon’s experience [69]. Another complication may arise if retroperitoneal lymph nodes are very close or adherent to a blood vessel, sometimes forcing the removal of a part of the vessel. While the obturator vessels may be dissected, the obturator nerve must be spared [48].

Damage to the obturator nerve may occur if it is clipped or otherwise injured, causing motor (adduction) and sensory impairments (in medial thigh). Dealing with this consequence requires intensive physiotherapy, vitamin B6 and pain reliever administration [71]. Respecting the parietal pelvic fascia provides the best results in the preservation of the autonomic pelvic nerves, since they are vital for the maintenance of the urogenital and anorectal functions [48]. Incidences of anatomic structures, pre-operatively present, have strong correlation with postoperative urine retention [69].

Attempts have been made to compare the risk of complications between different PLND extents. Several studies were held, producing conflicting results. Schwerfeld-Bohr et al., observed that lymphoceles developed more frequently in patients who underwent ePLND (17%) in comparison to lPLND (8%) [74]. These findings are supported by another randomized controlled trial (RCT), where ePLND and lPLND were performed at the same time, on the right and left hemi-pelvis, respectively. Lymphoceles and lower extremity edema were reported more often on the ePLND-performed side [75]. Fossati et al., analyzed data from 15 retrospective studies, discovering that some showed significant spikes in intra- and postoperative complications for ePLND, while others claim the difference as not statistically significant. The same conclusion was reached on the matter of lymphocele presence [41]. One study showed differences in urinary continence and erectile function recovery to be insignificant [75]. Due to wilder usage of modified templates (first introduced by Walsh), over 90% of patients have potency-preserving and nerve-sparing results [45,76]. Similarly, a larger surgical template paired with a worse preoperative state may explain the higher risk of complications in patients who underwent ePLND [34,77,78].

## 7. Surgical Techniques of PLND

Nowadays, there are three major types of surgical approaches to RP: open retro-pubic radical prostatectomy (RRP), laparoscopic RP, and, rapidly gaining popularity all over the world, robot-assisted radical prostatectomy (RARP). PLND can be effectively performed using any of these procedures. The greatest advantage of laparoscopic and robot-assisted techniques over open surgery lies is the decrease in peri- and postoperative complications [79]. Yuh et al., established that PLND can be safely and efficiently performed during RARP, even with increased nodal yield (>20) accompanied by ePLND [80]. In the study by Liss et al., it was confirmed that robotic ePLND presence enhanced detection of nodal metastases without an increase in complication rates [81]. This further addresses safety issues, while pointing a small but noticeable improvement in oncological outcome [82]. Overall, it is a matter of discussion whether perioperative complications and nodal yield are based on the individual qualities of a surgeon or are associated with surgical approach [83,84].

It is worth mentioning that in 2013 Mattei et al., presented a novel surgical procedure of PLND during RARP—the mono-block technique, which aims to be simple, while obtaining a good operative field control and resulting in radical removal of lymphatic tissue [85]. In a related study, an alteration of this method has been proposed: the “five-step monoblock sePLND” technique, which may be efficiently conducted when a risk of LNI ≥ 30% or nodal involvement is proved by MRI [86].

De Barros et al., have recently investigated PSMA-radio-guided surgery (RGS) translation to an RARP environment. Intraoperative detection of radiated nodes was carried out using the DROP-IN gamma probe. The research revealed sensitivity at the level of 86% and specificity at the level of 100%, with only one patient (1/20, 5%) suffering from a Clavien–Dindo grade > III complication [87].

## 8. New Perspectives for PLND

The most promising innovation, though still being tested and considered as an experimental therapy, is sentinel lymph node dissection (SLND). At present, PLND performed during prostatectomy remains the gold standard for nodal staging in prostate cancer, despite the fact that the rate of post-LND complications, as well as morbidity, rises as the number of dissected LNs grows [70,88]. Therefore, SLND, already a first choice procedure in melanoma, breast and penile cancers, is taken into consideration as an alternative [89]. According to the concept of metastatic spread of the tumor along the lymph drainage pathways, an assumption can be made that the absence of cancer invasion in the sentinel nodes is coequal with the lack of metastasis in other LNs [90]. The implementation of SLND would ideally prevent patients with PCa from overtreatment them with ePLND [91].

Sentinel node navigation surgery (SNNS) is employed to find out if there is a need for radical surgery and determine its extent in order to perform the least invasive procedure possible. In PCa, it is not a standard clinical procedure, and is mostly performed in clinical trials. A promising technique seems to be radioisotope-guided laparoscopic and robotic sentinel lymph node dissection [92,93,94].

The SN concept has poor value for regional metastases in prostate cancer, as recent findings imply the existence of a multitude of primary landing sites, calling into question whether isotope-based imagining accurately represents the nodal status of the entire pelvic basin, thus explaining the high false-negative rate of this procedure [33,37,95]. In high-risk PCa patients, Weckermann et al., presented that, in a group of 228 men, had only SN been removed, then ~⅓ of nodal metastases would have remained [38]. These drawbacks and the high false-negative rate for the detection of metastatic nodes are the major reasons why the sentinel node technique has not gained wide acceptance. However, according to Wawroschek et al., and Egawa et al., a patient is more likely to be LN-negative if he is SN metastasis-free [34,95].

The concept of SLND may carry many advantages, including a more tailored and balanced approach. Moreover, techniques used to detect SLN revealed lymph drainage pathways that have not been investigated before [70]. According to the collected data, some of the LNs on SLND may be observed outside of the ePLND template, and a few of these nodes may occur as positive [70,96]. Some studies show that SLND and ePLND have equal predictive value in the identification of metastatic lymph nodes. Adding SLND to ePLND improves BCR-free outcomes compared with ePLND only [89].

The SLND concept is unfortunately limited by current technical determinants, which are unable to properly detect all metastatic lymph nodes and the problem of only using intraoperative, rather than preoperative, methods of imaging SLN. Furthermore, the technique is bound by experimental protocols and a lack of standardized procedure guidelines.

Currently, the most frequently used methods of imaging SLN are radio-isotope injection of Technetium-99m and the indo-cyanine green (ICG) technique. The radioisotope SLN technique employs the trans-rectal injection of ^99m^Tc bound to a pharmaceutical in a prostate. Then, preoperative lympho-scintigraphy and SPECT-CT are performed, which is a valuable advantage for the creation of the surgical plan. Additional intraoperative usage of gamma-ray detection probes or gamma cameras enable the detection of sentinel lymph nodes and can provide the urologist with the precise location while performing lymphadenectomy [97]. Meta-analyses evaluating the detection of LN metastases suggested sensitivity of approximately 95% [7]. Another study, the first sentinel nomogram, shows a high degree of accuracy at the level of 82% and may be the first to aid clinicians in making a decision as to whether to implement SLND or choose conservative solutions [98]. The fluorescence imaging technique using ICG is based on the intraoperative detection of SLN with polarized light. Several studies have been reported and the median intraoperative SLN detection rates ranged between 76 and 97% [7]. Despite these promising results, ICG is still an unreliable SLN imaging method due to its poor diagnostic accuracy among patients with intermediate and high-risk PCa. The usage of hybrid techniques engaging both fluorescence and radioisotope methods is worth considering, the combination of which improves the detection rate.

Most recent diagnostic methods used in PCa concentrate on detecting metastatic and sentinel LN with higher predictive value and sensitivity. Doughton et al., investigated the first-in-human usage of ^68^Ga-Nanocolloid as the radiotracer for PET/CT lympho-scintigraphy and its results were promising, though this technique of SLN imaging requires further research. What is interesting is discovering unexpected lymph drainage patterns, including pathways leading to peri-vesicular, meso-rectal, inguinal and Virchow nodes [99]. Another innovative technique engages the prostate-specific membrane antigen—PSMA-labeled radiotracer (111In-PSMA-I&T). Maurer et al., were able to detect metastasis in LNs unrevealed by the ^68^Ga-PSMA PET method. One of the most promising techniques seems to be the usage of super-magnetic iron oxide (SPIO) nanoparticles as magnetic tracer for the MRI procedure proposed by Winter et al. [7].

## 9. Current Guidelines

Despite the growing body of evidence supporting the use of PLND during radical prostatectomy (RP), consensus regarding optimal management is distinctly absent. Guidelines for PLND in prostatectomy provided by the EAU indicate lymphadenectomy as the only available procedure for nodal staging, though failing to improve oncological outcomes. The EAU suggests the engagement of pre-operative tools predicting LNI in individual cases to avoid over-treating patients at low risk of nodal metastasis with PLND. Probability of LNI should be evaluated using either Briganti nomogram or Roach formula, as well as Partin and MSKCC nomograms; in cases of both of these tools, a risk of nodal invasion exceeding 5% should be considered a cut-off point above which ePLND is advisable [5].

The American Urological Association (AUA) suggests consideration of PLND for any localized PCa of intermediate risk or high risk. Patients should always be informed about the benefits of the procedure as well as the possible common complications, such as lymphocele. The guidelines also explain that evidence is lacking as to whether the removal of LNs containing metastatic prostate cancer has therapeutic benefits [100]. Additionally, AUA provides instructions concerning sub-stratification of patients. Thus, LND should be recommended especially for patients with unfavorable intermediate-risk or high-risk cohorts. The intermediate-risk subgroup categorization is based on Gleason score (3 + 4 being favorable and 4 + 3 being unfavorable) [100]. Furthermore, low-risk patients are also divided into very low-risk and low-risk groups, which is based on the number of biopsy cores and PSA density. AUA recommendations stay consistent with the NCCN guidelines [101].The National Institute for Health and Care Excellence (NICE) proposes RP as one of the treatment options for people with low-, intermediate- and high-risk localized PCa, and in said guidelines the procedures of PR and PLND are unseparated, which indicates that the lymphadenectomy of some extension should always be performed during PR. High-risk localized PCa is defined by PSA level (over 20 ng/mL), Gleason score (8 to 10), or clinical stage (T2c or more), and in this case RP should be taken into consideration when it is likely that the patient’s outcome can be controlled in the long term [102]. Table 1 briefly summarizes the most important aspects of LND indications, assessment of LNI probability, and extents of LND templates mentioned in the most popular guidelines.

## 10. Conclusions

Despite the growing body of evidence, several questions related to the role of pelvic lymphadenectomy in prostate cancer are unanswered. The progress in diagnostic techniques is significant, but we have not yet obtained satisfactory sensitivity and specificity levels, especially in the detection of nodal micro-metastases.

As a result, extended pelvic lymphadenectomy remains the gold standard in nodal staging with a possible positive oncological effect. However, it should be kept in mind that this procedure is not complication-free, even in the case of using new minimally invasive surgical techniques. Thus, patients should be carefully selected using the available guidelines and predictive tools. Doubts regarding the ideal anatomical template and oncological effectiveness clearly indicate the need for robust and high-quality clinical trials. Their results could allow for the formation of unambiguous clinical guidelines necessary for proper disease management.

## Figures and Tables

**Figure 1 jcm-11-02343-f001:**
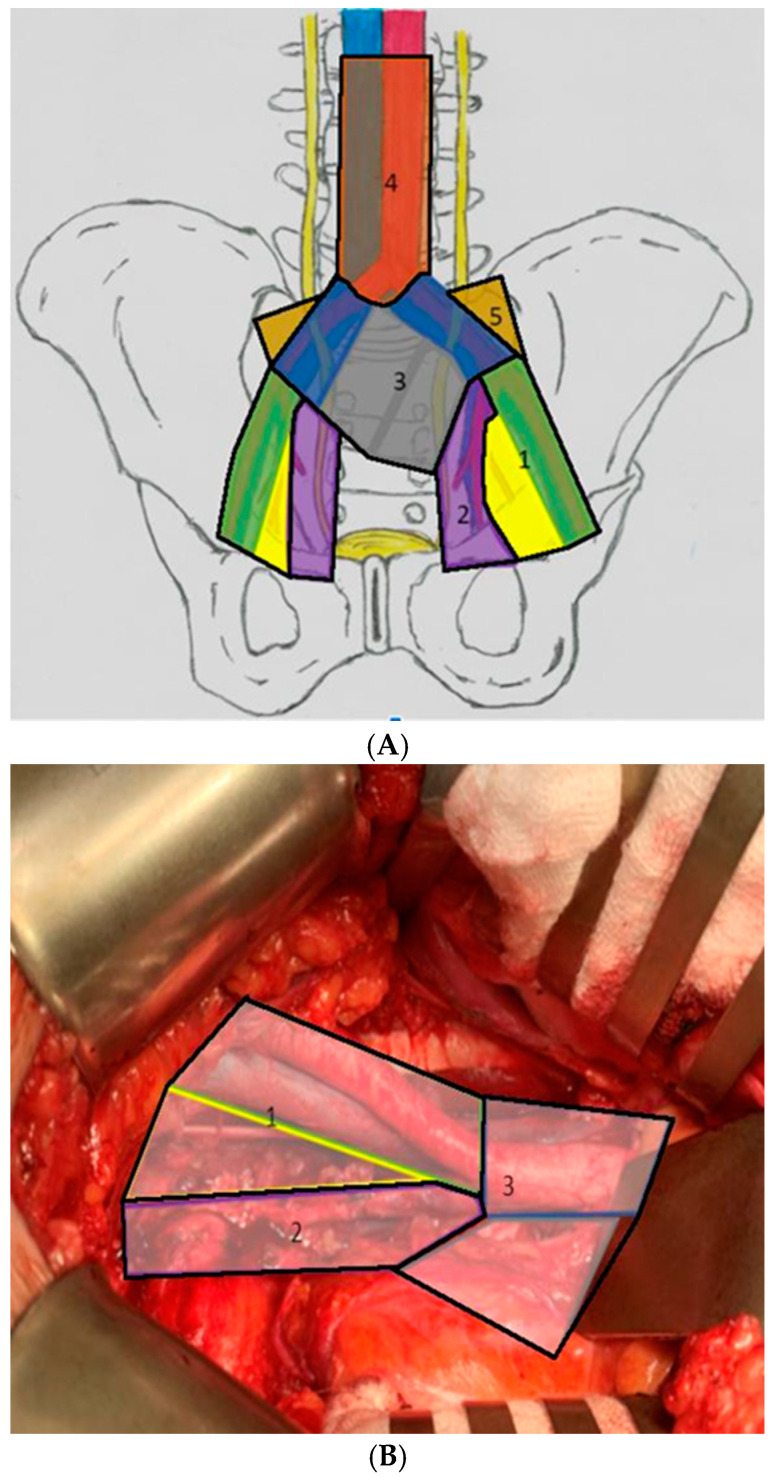
Anatomical extent of lymphadenectomy: 1—limited; 2—standard; 3—extended; 4—super-extended; 5—Marcille’s Fossa; (**A**)—topography; (**B**)—anatomical superimposing.

**Table 1 jcm-11-02343-t001:** Overview of indications and extents of LND in PCa according to guidelines provided by the EAU, the AUA, and the NICE.

Guidelines	Indications and the Extent of LND
EAU	- LND is indicated to be performed, when a risk of nodal invasion exceeds 5%. - Suggested nomograms assessing nodal involvement include the Briganti nomogram, the Roach formula, Partin tables and the MSKCC nomogram. - Extent: ePLND.
AUA	- LND should be considered for any localized PCa patients and should be recommended for unfavorable intermediate-risk and high-risk patients. - Nomograms assessing nodal involvement are briefly mentioned. - Extent: not specified, favorably ePLND.
NICE	- LND is indicated as a coherent part of RP. - The Roach formula is a recommended nomogram, though it is mentioned as a nodal involvement predictor in the section concerning RT usage in locally advanced PCa. - Extent: not mentioned.

EAU: The European Association of Urology; AUA: the American Urological Association; NICE: National Institute for Health and Care Excellence; LND: lymph node dissection; MSKCC: Memorial Sloan Kettering Cancer Center; ePLND: extended pelvic lymph node dissection; PCa: prostate cancer; RP: radical prostatectomy; RT: radiotherapy.

## Data Availability

Not applicable.

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
