# Peer review of "Utility of Lymphadenectomy in Prostate Cancer: Where Do We Stand?"

_jcm, 2022, doi:10.3390/jcm11092343_

Round 1
Reviewer 1 Report
Dear Authors,
The manuscript submitted by Małkiewicz B. et al. summarizes the current knowledge regarding the role of lymphadenectomy in prostate cancer. The review is interesting. The need for pelvic lymph node dissection in conjunction with radical prostatectomy is determined by the risk of regional lymph node involvement. The extent of a pelvic lymph node dissection impacts the frequency with which positive lymph nodes are detected and the incidence of complications. The optimal extent of pelvic lymph node dissection is uncertain, but the available evidence suggests that an extended dissection is preferable, especially in patients with very high-risk diseases.
The article is well written and organized.
Major aspects:
- Please include risk stratification provided by the National Comprehensive Cancer Network (NCCN), consistent with guidelines from the American Urological Association.
- Please include a table to summarize the indications and extensions of PLND in PCa according to the EAU, AUA, and NICE for a better understanding.
- Include a subsection regarding the type, complications, and oncological outcomes of PLND (open, laparoscopic, or robotic).
Minor aspects:
- l. 34-36 reference 5 is not adequate; please include guidelines
- l. 180-184 needs a reference
- Please include a list of abbreviations in alphabetical order
- The following article should be cited - https://doi.org/10.1016/j.urolonc.2018.11.020
Author Response
jcm-1656921
Response letter to the Reviewer #1 Report
We thank the Reviewer for encouraging feedback and appreciate the insightful comments and suggestions.
Below, we provide a point-by-point response to each of the reviewer’s comments.
All changes in the manuscript were highlighted in yellow for clarity.
We hope that the introduced revisions significantly improve the quality of this review and qualify it for further editorial stages.
Sincerely,
Authors
Major aspects:
- Please include risk stratification provided by the National Comprehensive Cancer Network (NCCN), consistent with guidelines from the American Urological Association.
Response: Thank you for this valuable suggestion. Indeed, substratification of patients (especially in the intermediate-risk group) is crucial. This information is included in the text – Section 8. Current guidelines.
- Please include a table to summarize the indications and extensions of PLND in PCa according to the EAU, AUA, and NICE for a better understanding.
Response: Table summarizing indications and extensions of PLND in PCa of different guidelines has been added to the manuscript.
- Include a subsection regarding the type, complications, and oncological outcomes of PLND (open, laparoscopic, or robotic).
Response: Thank you very much for this valuable comment, especially when minimally invasive techniques have developed. We have created a new paragraph dedicated to various surgical techniques - Section 6. Surgical techniques of PLND.
Minor aspects:
- 34-36 reference 5 is not adequate; please include guidelines
Response: Thank you very much for pointing out this inaccuracy. Proper citation was added.
- 180-184 needs a reference
Response: Thank you for your suggestion. We supplemented the bibliography with an article of the greatest level of relevance.
- Please include a list of abbreviations in alphabetical order
Response: This is great advice, considering the volume of the article and the multitude of different abbreviations used in the text. Thank you very much for pointing this out.
- The following article should be cited - https://doi.org/10.1016/j.urolonc.2018.11.020
Response: Thank you. Indeed, this article is very interesting. We have cited it in section 3 – Anatomical extent of PLND.
Reviewer 2 Report
Dear Colleagues,
“Utility of lymphadenectomy in prostate cancer: wehre do we stand?” by Malkiewicz et al. is an interesting review revolving around current applications and future perspectives of lymph nodes assessment in prostate cancer management and treatment.
The manuscript widely discusses imaging, surgical and oncological implications of aforementioned determinations; data explanation and articles reviewed are easy to read and understand, well-presented and well-discussed.
While the manuscript itself is quite interesting and informing, English could be improved.
The article itself reached the inner aim of the study, depicting the state of the art and the possible future perspective and implications of lymph nodes dissection; each paragraph was well-structured, consisting in a brief “introduction” section and a wide inner “discussion”.
The citations and analyses of other manuscripts are pleasing to read; each one is well-explained and fits the overall topic. It was nonetheless interesting to dive into their thesis in a trimodal structure of content presentation (radiology, surgery, oncology). I particularly enjoyed how Guidelines, scores and patients’ classifications were cited throughout the text.
While the structure is indeed promising, the conclusions could have been a little more elaborated; its core content is, regardless, punctual: as of today, we are in desperate need of new, more inclusive guidelines in which inform clinicians and oncology-revolving specialist more in general on the correct use of new sadiation strategies and interventions. "
Kind Regards
Author Response
jcm-1656921
Response letter to the Reviewer #2 Report
We thank the Reviewer for encouraging feedback and appreciate the insightful comments and suggestions.
We are very pleased that the reading of the article was a pleasure for the reader and allowed me to get involved in the meanders of PLND.
As recommended, the entire text has been checked and corrected by a native speaker from the MDPI.
As suggested by the reviewer, we have revised the Conclusions section. The paragraph has been redrafted and now contains unambiguous conclusions resulting from the literature review.
All changes in the manuscript were highlighted in yellow for clarity.
We hope that the introduced revisions significantly improve the quality of this review and qualify it for further editorial stages.
Sincerely,
Authors